# See Further When Clear: Adaptive Generative Modeling with Curriculum Consistency Model

## Abstract

Significant advances have been made in the sampling efficiency of diffusion models, driven by Consistency Distillation (CD), which trains a student model to mimic the output of a teacher model at an earlier timestep. However, we found that the learning complexity of the student model varies significantly across different timesteps, leading to suboptimal performance in consistency models. To address this issue, we propose the Curriculum Consistency Model (CCM), which stabilizes and balances the learning complexity across timesteps. We define the distillation process as a curriculum and introduce Peak Signal-to-Noise Ratio (PSNR) as a metric to quantify the difficulty of each step in this curriculum. By incorporating adversarial losses, our method achieves competitive single-step sampling Fréchet Inception Distance (FID) scores of 1.64 on CIFAR-10 and 2.18 on ImageNet 64x64. Moreover, our approach generalizes well to both Flow Matching models and diffusion models. We have extended our method to large-scale text-to-image models, including Stable Diffusion XL and Stable Diffusion 3.

## 1 Introduction

The development of generative models has become a prominent research focus in the field of deep learning. Variational autoencoders (VAEs) Kingma (2013) are preferred for their ease of training, but they frequently experience posterior collapse during image generation, leading to blurred results. Generative adversarial networks (GANs) Goodfellow et al. (2014), in contrast, can generate high-quality images, but the instability of their training process remains a significant challenge. Recently, diffusion models Ho et al. (2020),Song et al. (2020),Song et al. (2021) have received attention for their ability to produce high-quality images, but despite this, their performance in sampling efficiency is not satisfactory and often requires a lot of functional evaluation. Compared to diffusion models, Flow Matching (FM) Lipman et al. (2023) is a simulation-free method with more deterministic trajectories, making it a potentially more robust and stable alternative for training generative models. However, FM still requires multiple function evaluations to generate high-quality images.

With the introduction of the Consistency Models (CM) Song et al. (2023), researchers have shifted their focus to distillation methods to enhance sampling efficiency. CM constrains any point on the trajectory to the same solution by self-consistency, thus reducing the number of function evaluations. Latent diffusion models (LCM) Rombach et al. (2022) use consistency constraints in the latent space and extend the models to high-resolution text-to-image synthes. Consistency Trajectory Models (CTM) Kim et al. (2023) further allows unlimited traversal along the probability flow Ordinary Differential Equations (ODEs) between arbitrary starting and ending points during diffusion process.

As shown in Figure 1a, at timestep $t$ (where $t \in [0, 1)$, a common approach in Consistency Distillation (CD) is to encourage the student model to mimic the output of the teacher model at timestep $u$ (where $u \in (t, 1]$). However, we found that the learning complexity of the student model is highly unstable across different timesteps, leading to unsatisfactory semantic structure and poor text-image alignment in the consistency model. We analyze learning complexity in Figure 2, where we quantify the learning complexity by calculating the Peak Signal-to-Noise Ratio (PSNR) between the student and teacher outputs at different timesteps. The results indicate that the learning complexity for student model increases gradually as $t$ progresses from smaller values (corresponding to near-

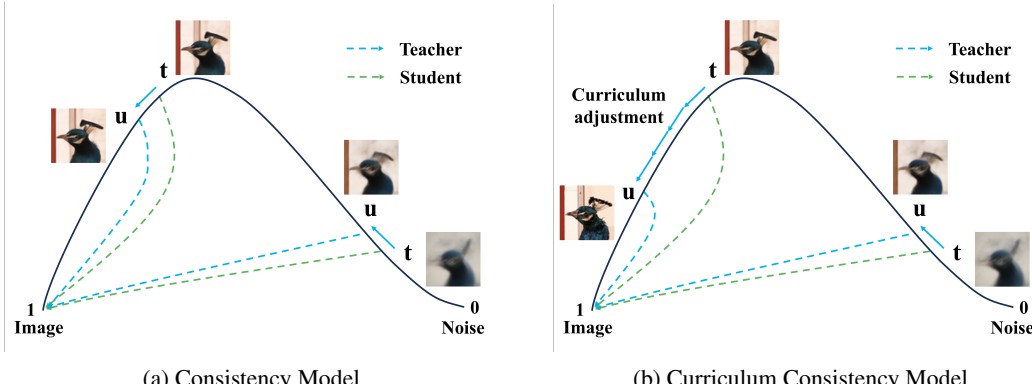

(a) Consistency Model  (b) Curriculum Consistency Model

Figure 1: Comparison between Consistency Model (CM) and Curriculum Consistency Model (CCM). CM encourages the student model to mimic the output of the teacher model, on this basis, CCM further guide the student model to learn at the more challenging timesteps.

pure noise) to larger values (closer to the final image). However, most studies Song et al. (2023), Luo et al. (2023) suffer from the instability of learning complexity, as they sample uniformly along the timesteps and use a fixed timestep size for the consistency distillation. As a result, the student model struggles to learn effectively at the more challenging timesteps (when $t \to 0$), which are more closely related to the semantic generation in the diffusion model as shown in Figure 1b.

To address these issues, we propose an adaptive training method that stabilizes and balances the learning complexity of the model under varying noise intensities, as shown in Figure 1b. We first measure the learnning complexity of the current training step. Then, Our approach integrates curriculum learning into the distillation process to dynamicly adjust the difficulty of the learning targets. Specifically, we use PSNR to quantify the difficulty and dynamically modify the learning objectives. To ensure high-quality teacher outputs, we efficiently adopt a multi-step iterative generation strategy.

In summary, we propose Curriculum Consistency Model (CCM) to perform the consistency distillation for the diffusion model. Our main contributions are as follows:

- We identify the instability in learning complexity during consistency distillation, which significantly impacts text-to-image alignment and the generation of semantic structures in diffusion models.
- We introduce PSNR to assess curriculum difficulty and design a more effective adaptive noise schedule to maintain curriculum consistency across different training samples.
- By incorporating adversarial losses, our method achieves high-quality few-step generation. Specifically, we obtain one-step sampling Fréchet Inception Distance (FID) scores of 1.64 on CIFAR-10 and 2.18 on ImageNet 64x64.
- We extend our method to large-scale high-resolution image generation models including Stable Diffusion XL Podell et al. (2024) and Stable Diffusion 3 Esser et al. (2024). Our results show that the introduction of curriculum consistency leads to lower FID, higher CLIP scores, and significantly improved semantic understanding in the generated images.

## 2 PRELIMINARIES

### 2.1 FLOW MATCHING WITH OPTIMAL TRANSPORT

Given a data space $\mathbb{R}^d$ with data points $x = (x^1, ..., x^d) \in \mathbb{R}^d$, we can define a time-dependent probability density $p_t(x)$ and a vector field $u_t(x)$. The flow $\psi_t(x)$, which is a time-dependent diffeomorphic map induced by $u_t(x)$, can be derived using the ordinary differential equation (ODE):

$$\frac{d}{dt}\psi_t(x) = u_t(\psi_t(x)), \quad \psi_0(x_0) = x_0 \tag{1}$$

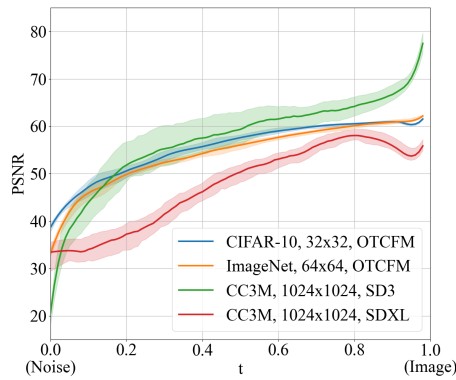 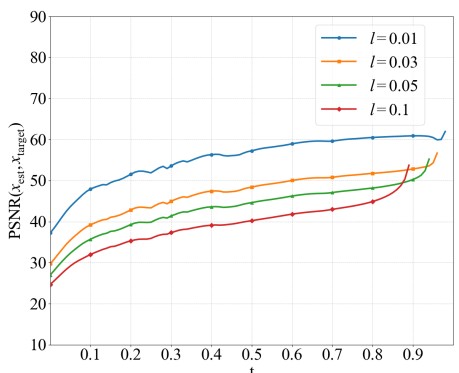

Figure 2: Learning Complexity Investigation: Analysis of the PSNR between the student and teacher model outputs across different timesteps on various datasets for both Flow Matching models and diffusion models.

Figure 3: The relationship of PSNR with different distillation step $l$. The learning complexity remains consistent across various distillation steps.

By modeling the vector field $u_t(x)$ with a neural network $v_t(x; \theta)$, we obtain a Continuous Normalizing Flow (CNF) that transforms a density $p_0$ to $p_1$ via the push-forward equation Chen et al. (2018):

$$p_t(x) = p_0(\psi_t^{-1}(x)) \left| \det \left[ \frac{\partial \psi_t^{-1}}{\partial x}(x) \right] \right| \tag{2}$$

A vector field $u_t(x)$ is said to generate a probability path $p_t(x)$ if its flow $\psi_t(x)$ satisfies Eq.2.

Flow Matching (FM) Lipman et al. (2023) is a simulation-free method for training CNFs by regressing onto a target vector field $u_t(x)$. They derive a simplified objective of Conditional Flow Matching (CFM) with $x_t = \psi_t(x_0 | x_1)$:

$$\mathbb{E}_{t, q_1(x_1), p_t(x|x_1)} \| v_t(x; \theta) - u_t(x_t | x_1) \|^2 \tag{3}$$

A specific choice of $p_t(x|x_1)$ is the optimal transport displacement interpolant and the corresponding vector field is defined as:

$$u_t(x_t | x_1) = \frac{x_1 - x}{1 - t} \tag{4}$$

where $x_t = \psi_t(x_0|x_1) = (1-t)x_0 + tx_1$. This results in straight paths from $x_0$ at $t = 0$ to $x_1$ at $t = 1$, known as stochastic interpolants. This approach generalizes to Gaussian conditional paths, where $p_t(x|x_1) = \mathcal{N}(x|\mu_t(x_1), \sigma_t(x_1)^2 I)$, encompassing most prior diffusion models.

The objective of CD is to align the neural mapping $G_\theta$ with the true mapping $G$ by ensuring $G_\theta(x_t, t, 1) \approx G(x_t, t, 1), \forall t \in [0, 1)$. We train $G_\theta$ by comparing it with the numerical solution of the pre-trained Probability Flow ODE (PF ODE) solver,

$$G_\theta(x_t, t, 1) \approx \text{Solver}(x_t, t, 1; \phi) \approx G(x_t, t, 1) \tag{5}$$

where $\phi$ means the teacher model. To simplify the training process, we adopt a local consistency matching approach. Specifically, we compare the student's prediction with the result obtained by solving the PF ODE over the interval $(t, u)$ using the teacher model, followed by mapping to time 1 using the target model:

$$G_\theta(x_t, t, 1) \approx G_{\theta^-}(\text{Solver}(x_t, t, u; \phi), u, 1) \tag{6}$$

where $u$ is randomly sampled from $(t, 1)$, and $\theta^-$ denotes the exponential moving average (EMA) of the parameters, $\theta^- \leftarrow stopgrad(\mu\theta^- + (1 - \mu)\theta)$. This method ensures that the student model effectively distills information from the teacher model over the interval $(t, u)$.

## 2.2 Consistency Models and Consistency Distillation

The inverse of the diffusion process can be represented by a deterministic Probability Flow ODE (PF ODE) which is given by Song et al. (2021) :

$$dx = \left[-\tfrac{1}{2}\beta_\sigma x_\sigma - \tfrac{1}{2}\beta_\sigma \mathbf{s}_\theta(x_\sigma, \sigma)\right] d\sigma \tag{7}$$

where $\sigma$ means signal-to-noise ratio. Consistency models aim to simplify multiple evaluations of $\mathbf{s}_\theta(x, \sigma)$ by directly learning an ODE that maps any point $(x_\sigma, \sigma)$ on a trajectory to $x_\epsilon$, where $\epsilon$ is a small positive value to ensure numerical stability Song et al. (2023). The consistency function $f_\theta$ is defined as:

$$f : (x_\sigma, \sigma) \mapsto x_\epsilon, \sigma \in [\epsilon, T] \tag{8}$$

A common implementation of the consistency function involves a skip connection structure:

$$f_\theta(x, \sigma) = c_{\text{skip},\sigma}x + c_{\text{out},\sigma}F_\theta(x, \sigma) \tag{9}$$

where $c_{\text{skip},\epsilon} = 1$ and $c_{\text{out},\epsilon} = 0$, ensuring that $f(x_\epsilon, \epsilon) = x_\epsilon$. The generation process begins by sampling $x_T \sim p_T(x_T)$, and then directly obtaining $x_\epsilon$ through $f_\theta(x_T, T)$. The direct optimization objective is:

$$\|f_\theta(x_\sigma, \sigma) - x_\epsilon\|^2 \tag{10}$$

A practical solution is to enforce consistency between two adjacent points on the trajectory. By discretizing the interval $[\epsilon, T]$ into $N$ steps, $\sigma_i = \left(\epsilon^{1/\rho} + \frac{i-1}{N-1}(T^{1/\rho} - \epsilon^{1/\rho})\right)^\rho$ Karras et al. (2022), we can approximate $\hat{x}_\phi(\sigma_n)$ using Euler's method, and the resulting loss function is:

$$\mathcal{L}_{\text{CD}}^N(\theta, \theta^-; \phi) = \mathbb{E}_{n \sim \mathcal{U}[1, N-1]}\left[\lambda(\sigma_n)d\left(f_\theta(x_{\sigma_{n+1}}, \sigma_{n+1}), f_{\theta^-}(\hat{x}_{\phi,\sigma_n}, \sigma_n)\right)\right] \tag{11}$$

where $\lambda(\sigma_n) = 1$ and $d(\cdot, \cdot)$ is a distance metrics.

## 3 Problem Analysis

In generative models based on denoising, the varying levels of noise in the input can lead to different signal-to-noise ratios (SNR) during the denoising process, as discussed in Karras et al. (2022); Hang et al. (2023). Consequently, at different training steps, the difficulty that generative models learn varies, which in turn affects the model's convergence rate and the quality of the generated results. The core of the learning complexity lies in the magnitude of the difference between the model's predicted results and the ground truth. Inspired by this phenomenon, we conducted an in-depth examination of the learning complexity during the consistent model learning process by comparing the outputs of the student model with those of the teacher model.

In this article, we propose using Peak Signal-to-Noise Ratio (PSNR) to access learning complexity, as PSNR is widely used to measure the difference between a denoised image and its original counterpart. Specifically, given the outputs of the student model, $x_{\text{target}}$, and those of the teacher model, $x_{\text{est}}$, PSNR is calculated using the following formula:

$$\delta = 10 \log_{10}\left(\frac{\text{MAX}^2}{\text{MSE}(x_{\text{est}}, x_{\text{target}})}\right). \tag{12}$$

A high PSNR means little difference between $x_{\text{target}}$ and $x_{\text{est}}$ with low learning complexity, and vice versa.

We conduct experiments on both diffusion-based and FM-based models (SDXL Podell et al. (2024), SD3 Esser et al. (2024), OTCFM Tong et al. (2023)) and select 3 classic datasets (CIFAR-10, ImageNet, and CC3M) covering both low and high resolutions (32x32, 64x64, and 1024x1024) to ensure reliability and robustness. The mean and variance of PSNR between the student and teacher model outputs on $t$ are shown in Figure 2. We observe that the PSNR value consistently increases as $t$ progresses from 0 to 1, indicating a gradual reduction in the model's learning complexity. This aligns with our intuition: when $t$ is near 0, the PSNR is typically around 30, as the input is heavily mixed with noise, leading to high learning complexity. At this stage, the model is prone to confusion, causing instability and slow convergence. Conversely, when $t$ approaches 1, the PSNR usually exceeds 50, indicating that the learning complexity is too low, resulting in reduced learning efficiency. We argue that this instability and inefficiency hinder the overall learning process of the CM model.

We further explore the effect of distillation step $l = u - t$, and present the results in Figure 3. The value of $l$ typically serves as a hyperparameter in the CM model, and greatly influences the effectiveness of the model's learning. In Figure 3, it can be observed that different values of $l$ yield consistent results across timesteps.

Can we mitigate this imbalance in learning complexity to enhance the effectiveness of CM learning? In this paper, we attempt to present a feasible solution by proposing an adaptive method named the Curriculum Consistency Model (CCM) which will be elaborated in the following section.

## 4 METHOD

### 4.1 CURRICULUM CONSISTENCY MODEL

Our goal is to design an algorithm that ensures a stable and balanced learning complexity for the model under different noise intensities and at various training iterations. To achieve this, we should see further when clear, thus, we propose the Curriculum Consistency Model (CCM). CCM incorporates three key designs, which are 1. A reliable metric for measuring the difficulty of learning, 2. Dynamic adjustment of learning objectives based on the difficulty of learning, and 3. Multi-step iterative generation to ensure the quality of learning objectives.

**Measuring the difficulty of learning.** We directly use PSNR to measure the learning complexity according to Eq. 12. We have shown the stability and generaliability of PSNR across different datasets, different noise intensity and different training periods in Section 3.

**Dynamic adjustment of learning objectives.** In order to adjust the learning complexity, we change the output of teacher model $x_{\text{target}}$. At each training step, we cycle between estimating the learning complexity and modifying $u$ until the learning complexity exceeds a certain fixed value. At different values of $t$ and during various training steps, we may obtain different values of $u$, showing the adaptive nature of CCM.

**Multi-step iterative generation.** There are various methods for generating $x_u$. A straightforward approach is to estimate $u$ directly from $t$ without regard for the magnitude of the difference between $u$ and $t$. However, CCM may select a $u$ that is significantly greater than $t$ to ensure a stable learning complexity, which could lead to the teacher model making inaccurate predictions due to a large timestep size. Consequently, this may result in the student model learning targets that are vague or inaccurate. Therefore, we propose a multi-step iterative generation method where the teacher model will iterate one step forward each time until the estimated model difficulty meets the requirements, which are currently unknown.

For clarity, we have written the CCM algorithm's procedure in pseudocode and presented it in Algo1.

### 4.2 IMPLEMENTATION

CCM focuses on addressing general issues in CM, thus making it applicable to a variety of common denoising-based generative models, including diffusion and flow matching.

---

**Algorithm 1** PSNR-Adjusted Target Computation

---

1: **Input:** noisy input $x_t$, timestep size $s$, condition $c$, threshold $T_{\text{PSNR}}$, teacher model $\phi$, target model $\theta^-$, student model $\theta$
2: **Output:** PSNR-Adjusted target $x_{\text{target}}$
3: Sample $t \sim \mathcal{U}(0,1)$
4: Calculate $v_t = G_\theta(t, x_t, c)$
5: Calculate $x_{\text{est}} = \text{Solver}(v_t, x_t, t, 1)$
6: **repeat**
7:     Calculate $v_t^\phi = G_\phi(t, x_t, c)$
8:     Update $u \leftarrow \min(t+s, 1)$
9:     Calculate $x_u = \text{Solver}(v_t^\phi, x_t, t, u)$
10:     Compute $v_u = G_{\theta^-}(u, x_u, c)$
11:     Compute $\hat{x}_1^u = \text{Solver}(v_u, x_u, u, 1)$
12:     Compute $\delta = \text{PSNR}(x_{\text{est}}, x_{\text{target}})$
13:     Update $t \leftarrow u, x_t \leftarrow x_u$
14: **until** $\delta < T_{\text{PSNR}}$ or $u == 1$

---

**CCM with diffusion models.** In diffusion models, we compute the target as

$$\mathcal{L}_{\text{CCM}}(\theta; \phi) := \mathbb{E}_{\sigma \in [\epsilon, T]} \mathbb{E}_{\tau \in [\epsilon, \sigma)} \mathbb{E}_{x_\epsilon} \mathbb{E}_{x_\sigma | x_\epsilon} [d(f_{\theta^-}(\text{Solver}(x_\tau, \tau, \sigma), \tau; \phi), f_\theta(x_\sigma, \sigma)]. \quad (13)$$

**CCM with flow match models.** In flow matching models, we compute the target as

$$\mathcal{L}_{\text{CCM}}(\theta; \phi) := \mathbb{E}_{t \in [0,1)} \mathbb{E}_{u \in (t,1]} \mathbb{E}_{x_1} \mathbb{E}_{x_t | x_1} [d(G_{\theta^-}(\text{Solver}(x_t, t, u; \phi), u, 1), G_\theta(x_t, t, 1)]. \quad (14)$$

### 4.3 ADVERSARIAL LOSSES

In generative modeling, student models derived from distillation often produce lower-quality samples compared to their teacher models, as they rely solely on distillation losses. To improve the student's performance and potentially surpass the teacher in quality, we incorporate adversarial training into our framework. Previous work, such as Esser et al. (2021) and Kim et al. (2023), has demonstrated that combining reconstruction and adversarial losses significantly enhances image generation quality.

Our Curriculum Consistency Model (CCM) framework integrates both PSNR-adjusted distillation loss and adversarial losses into a unified training objective:

$$\mathcal{L}_{\text{GAN}}(\theta, \eta) = \mathbb{E}_{x_1}(\log d_\eta(x_1) + \mathbb{E}_{t \in [0,1)} \mathbb{E}_{x_1} \mathbb{E}_{x_t | x_1} [\log(1 - d_\eta(x_{\text{est}}(x_t, t, 1)]  \quad (15)$$

$$\min_\theta \max_\eta \mathcal{L}(\theta, \eta) = \mathcal{L}_{\text{CCM}}(\theta; \phi) + \lambda_{\text{GAN}} \mathcal{L}_{\text{GAN}}(\theta, \eta) \quad (16)$$

where $d_\eta$ represents the discriminator network and $\lambda_{\text{GAN}}$ is an adaptive weighting. Details are in Kim et al. (2023).

## 5 EXPERIMENTS

### 5.1 EXPERIMENTAL DETAILS

**Datasets**. For low-resolution image generation, we train models on CIFAR-10 Krizhevsky et al. (2009) and ImageNet 64x64 Deng et al. (2009) datasets and evaluate on the same datasets. For high-resolution image generation, we train LoRA weights Hu et al. (2022) on the CC3M Changpinyo et al. (2021) dataset and evaluate on COCO-2017 Lin et al. (2014) with our chosen 5K split.

**Models**. We verify the image generation based on both flow match and diffusion models, including Optimal Transport Conditional Flow Matching (OT-CFM) Tong et al. (2023), Stable Diffusion 3 Esser et al. (2024), and Stable Diffusion XL Podell et al. (2024). Our code implementation is based on torchcfm and phased consistency model Wang et al. (2024).

**Evaluation Metrics**. We report the FID Heusel et al. (2017) and CLIP score Radford et al. (2021) of the generated images and the validation 5K-sample splits.

Our experimental parameters are shown in Appendix A.1.

## 5.2 EXPERIMENTAL RESULTS AND ANALYSIS

(a) Performance comparisons on CIFAR-10

| Model Type | Method | NFE (↓) | FID (↓) |
|---|---|---|---|
| GAN | StyleGAN-XL(Sauer et al. (2022)) | 1 | 1.85 |
| Diffusion | DDPM(Ho et al. (2020)) | 1000 | 3.17 |
| | DDIM(Song et al. (2020)) | 100 | 4.16 |
| | Score SDE(Song et al. (2021)) | 2000 | 2.20 |
| | EDM(Karras et al. (2022)) | 35 | 2.01 |
| | 2-Rectified Flow(Liu et al. (2023)) | 1 | 4.85 |
| | CD(Song et al. (2023)) | 1 | 3.55 |
| | CD + GAN(Lu et al. (2023)) | 1 | 2.65 |
| | CTM(Kim et al. (2023)) | 1 | 1.98 |
| Flow Match | OT-CFM(Tong et al. (2023)) | 100 | 6.29 |
| | PCM(Wang et al. (2024)) | 8 | 1.94 |
| | **CCM (ours)** | 1 | **1.64** |

(b) Performance comparisons on ImageNet 64×64

| Model Type | Method | NFE (↓) | FID (↓) |
|---|---|---|---|
| Diffusion | EDM(Karras et al. (2022)) | 79 | 2.44 |
| | CD(Song et al. (2023)) | 1 | 6.20 |
| | CTM(Kim et al. (2023)) | 1 | **1.92** |
| Flow Match | OT-CFM(retrained) | 100 | 5.36 |
| | **CCM (ours)** | 1 | 2.18 |

(c) Performance comparisons on CoCo2017-5K

| Base Model | Method | CLIP Score (↑) | Resized FID (↓) |
|---|---|---|---|
| SD3 | Original | 28.09 | 99.61 |
| | LCM(Wang et al. (2024)) | 32.32 | 35.62 |
| | PCM(Wang et al. (2024)) | **32.44** | 33.55 |
| | **CCM(ours)** | 32.42 | **32.54** |
| SDXL | Original | 30.41 | 70.28 |
| | Hyper-SD(Ren et al. (2024)) | 32.10 | 30.38 |
| | PCM(Wang et al. (2024)) | 32.47 | 29.89 |
| | **CCM(ours)** | **32.60** | **28.90** |

Table 1: Performance comparisons on different datasets.

Based on the experimental results provided in Table 1, we conduct a performance analysis of the Curriculum Consistency Model (CCM) compared to existing approaches. On the CIFAR-10 dataset, CCM achieves an impressive unconditional FID of 1.64 with only one function evaluation (NFE=1), outperforming other methods. CCM not only surpasses these methods in sampling efficiency but also achieves superior image quality. On the ImageNet 64×64 dataset, CCM also performed strongly: CCM's FID (NFE=1) reaches 2.18 on conditional generation, which is also competitive with the mainstream generated models. The samples generated by CCM (NFE=1) trained on CIFAR-10 and ImageNet 64x64 are shown in Figure 4. CCM show excellent acceleration that the images generated by CCM in one step are comparable in quality to those generated by OT-CFM in 100 steps, and at least 50x faster in inference. Additional images are provided in the appendix for further reference A. The training cost of CCM will be discussed in ablation studies.

When scaled to large scale methods and high resolution, CCM can still maintain advantages. According to Table 1(c), CCM has achieved lower FID on both FM and Diffusion-based methods, even the CLIP score on Diffusion-based methods has improved. We compare the samples generated by different methods and find that CCM performs better semantic comprehension (Figure. 5) and struc-

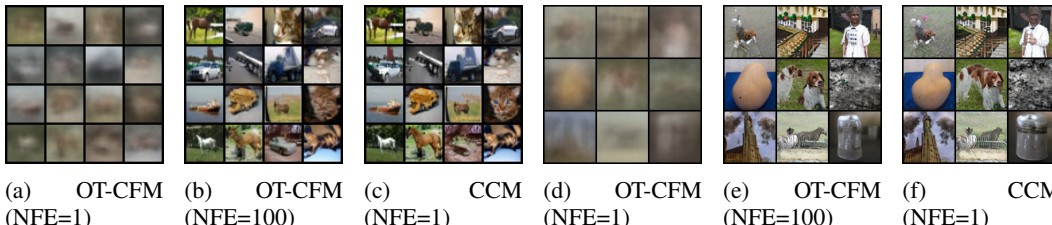

(a)      OT-CFM
(NFE=1)

(b)      OT-CFM
(NFE=100)

(c)            CCM
(NFE=1)

(d)      OT-CFM
(NFE=1)

(e)      OT-CFM
(NFE=100)

(f)              CCM
(NFE=1)

Figure 4: Samples generated by OT-CFM and CCM on CIFAR-10 and ImageNet 64x64.

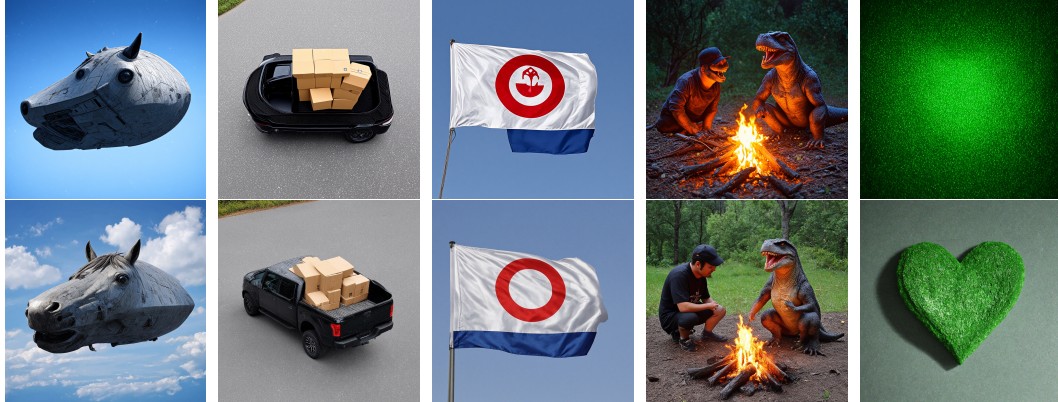

(a) A high-quality photo of a spaceship that looks like the head of a horse.

(b) an overhead view of a pickup truck with boxes in its flatbed.

(c) a white flag with a red circle next to a solid blue flag.

(d) Photo of a T-Rex wearing a cap sitting at a bonfire with his human friend.

(e) A green heart with shadow.

Figure 5: Semantic comparison of images generated by PCM (up) and CCM (down). CCM shows better semantic understanding and generates images that better fit the text.

tural rationality (Figure. 6). The results indicate that our method demonstrates strong generalization capabilities.

## 5.3 ABLATION STUDIES

We perform thorough ablation studies to evaluate the impact of different modules in the method. All ablation experiments are based on CIFAR-10, with adversarial training.

**Static vs. Dynamic.** We first compared different target selection strategies, namely static strategies and dynamic strategies. Static strategies include varying numbers of iterative generation steps and single-step timestep sizes $s$, while the dynamic strategy is CCM and inverse-CCM. From Table 2, we can observe that CCM surpasses all other strategies. Moreover, when the number of iterative steps increases from 1 to 3, the model's performance improves. Similarly, increasing the distillation step $l = u - t$ also exhibits a similar phenomenon, but a larger number of $l$ with less iterative steps can be detrimental. Furthermore, we experimented with varying the timestep size in accordance with the changes in $t$. Increasing $l$ proportionally as $t$ increases is not a good choice since it is almost impossible to learn when both $t$ and timestep size $s$ are very small, which also reminds us to balance learning complexity and model ability. A special case of the opposite is learning ground truth directly, i.e., $l = s = 1 - t$, which also lags behind CCM. Last, we compare the results of using different learning complexity metrics in dynamic methods and found that using inverse-CCM not only performs worse than CCM, but is also inferior to some static methods.

**Strategies of determining $x_{\text{target}}$** We tested various methods for determining $x_{\text{target}}$, including single-step iteration and multiple-steps with different timestep sizes $s$ in Table 3. The effect of directly generating $x_u$ from $x_t$ is poor compared to the effect of multi-step generation. This may

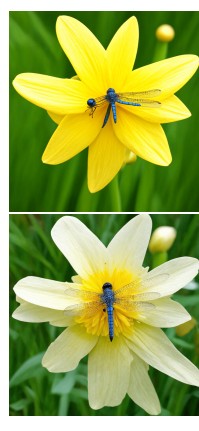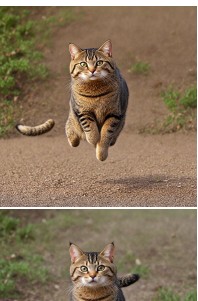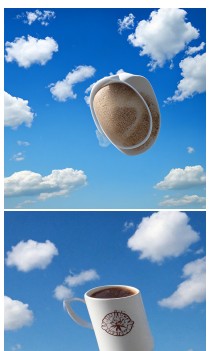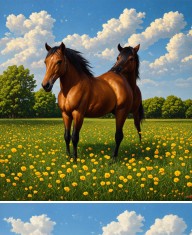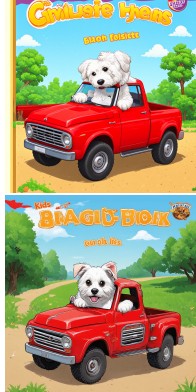

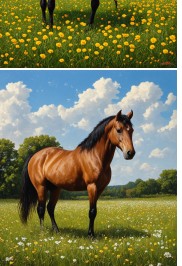

(a) a close-up of a blue dragonfly on a daffodil.  (b) a cat jumping in the air.  (c) a coffee mug floating in the sky.  (d) a dutch baroque painting of a horse in a field of flowers.  (e) a kids' book cover with an illustration of white dog driving a red pickup truck.

Figure 6: Structure comparison of images generated by PCM (up) and CCM (down). Both models correctly understand the text, but the structures generated by CCM are more reasonable.

| Strategy | Method | Steps | FID ($\downarrow$) |
|---|---|---|---|
| Static | $l = 0.01$ | 1 | 14.06 |
| | $l = 0.03$ | 1 | 11.38 |
| | $l = 0.1$ | 1 | 16.2 |
| | $l = 0.06$ | 2 | 10.15 |
| | $l = 0.09$ | 3 | 9.89 |
| Dynamic | $l = 0.1t$ | 1 | 27.19 |
| | inverse-CCM | - | 12.66 |
| | $l = 1 - t$ | 1 | 10.67 |
| | **CCM** | - | **9.32** |

Table 2: Comparison between static and dynamic strategies. For CCM, $T_{\text{PSNR}} = 40, s = 0.03$. Reverse-CCM adopts the opposite strategy of CCM.

| Method | Timestep size | FID ($\downarrow$) |
|---|---|---|
| Single-step | - | 46.82 |
| Multi-steps | $s = 0.01$ | 9.96 |
| | $s = 0.03$ | **9.32** |
| | $s = 0.05$ | 9.78 |

Table 3: Comparisons among strategies of determining $x_{\text{target}}$, $T_{\text{PSNR}} = 40$.

| $T_{\text{PSNR}}$ | FID ($\downarrow$) | Training (hour/100K) | Iteration (FID=12) |
|---|---|---|---|
| - | 11.84 | 10.45 | 80K |
| 35 | 10.26 | 20.97 | 25K |
| **40** | **9.32** | 18.81 | 32K |
| 45 | 9.95 | 18.19 | 35K |

Table 4: The choice of $T_{\text{PSNR}}$, $s = 0.03$.

be because the quality of the directly generated $x_u$ is relatively low, which affects the effectiveness of CM learning. We also found that after using CCM, the model is no longer sensitive to timestep sizes, with $s = 0.03$ slightly outperforming other choices.

**The choice of $T_{\text{PSNR}}$.** Different PSNR values determine the dynamically selected number of iterative steps during the training process, which is a hyperparameter in the methods presented in this paper. We conducted experiments with different values of $T_{\text{PSNR}}$, as shown in Table 4. It can be observed that the FID results first decrease and then increase as the threshold value increases. However, within the range of 35-45, the results are better than the baseline, indicating that our method is not very sensitive to PSNR. Moreover, although CCM will lead to an increase in the time of a single iteration, the convergence rate is accelerated at the same time. Based on the same FID, the CCM method accelerates by about 20% on average and achieves a lower FID, bringing significant benefits.

**Effect of GAN loss.** It can be seen from Figure 7 that either in vanilla distillation or in CCM, FID decreases significantly benefit from the ground truth indirectly introduced by adversarial training.

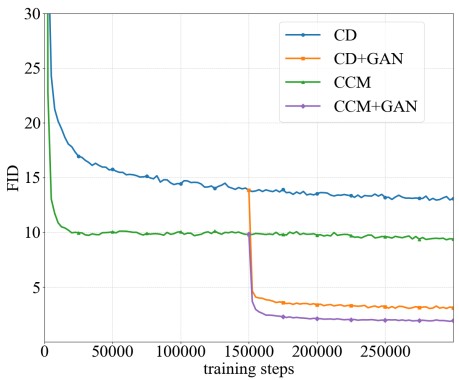

Figure 7: The effect of adversarial loss.

## 6 RELATED WORKS

**Diffusion model (DM)**. DMs have become a leading approach in high-fidelity image generation Rombach et al. (2022). Recent work focuses on improving sample quality Ho et al. (2020), optimizing density estimation Song et al. (2021), and accelerating the sampling process. Some studies explore the underlying mechanisms and design space of DMs, while others scale up DMs for text-conditioned image synthesis Podell et al. (2024) or improve sampling efficiency through methods in the latent spaceSong et al. (2020).

**Consistency models (CM)** The consistency model Song et al. (2023) represents a new family of generative models that can be trained either via distillation or without teacher models, often surpassing diffusion models in performance. The Consistency Trajectory Model (CTM) Kim et al. (2023) innovatively introduces trajectory consistency, offering a flexible framework for training. Recent multistep consistency models propose splitting ODE trajectories for improved consistency learning Wang et al. (2024).

**Flow Matching (FM)** Flow Matching (FM) learns a vector field that generates an ODE for a desired probability path, without requiring computationally intensive simulations Lipman et al. (2023). This flexibility has led to various efforts to improve trajectory properties, particularly straightness, which enables efficient simulation with fewer steps. Methods like Multisample FM Pooladian et al. (2023) and Minibatch OT Tong et al. (2023) aim to straighten trajectories through optimal transport plans, but these approaches are computationally prohibitive. Rectified Flow Liu et al. (2022) and Optimal FM offer alternatives.

## 7 CONCLUSION

In this article, we introduce the use of PSNR to measure the difficulty in the CM learning process and have discovered that the distribution of difficulty is highly imbalanced under different noise intensities. To alleviate this issue, we propose Curriculum Consistency Model (CCM), an efficient method for training models based on Neural Ordinary Differential Equations (ODEs). We design an adaptive noise schedule to maintain the consistency of curriculum difficulty and verify the rationality and validity of the design. By incorporating adversarial losses, our method achieves comparable single-step sampling Fréchet Inception Distance (FID) results on CIFAR-10 (1.64) and ImageNet64x64 (2.18). More importantly, our approach is not limited to FM, it works on diffusion models as well and we have successfully extended the proposed method to large-scale models, such as Stable Diffusion XL and Stable Diffusion 3. We hope that our paper will inspire greater attention to the issue of difficulty in the CM learning process and attract more researchers to engage in related research questions, such as dynamic PSNR thresholds, sampling probabilities of $t$, and so on.

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
