# A APPENDIX

## A.1 EXPERIMENTAL HYPERPARAMETERS

Table 5: Experimental details on hyperparameters.

| Hyperparameter | CIFAR-10 32x32 | ImageNet 64x64 | CC3M 1024x1024 |
|---|---|---|---|
| Training type | unconditional | conditional | conditional |
| Learning rate | 2e-4 | 1e-5 | 5e-6 |
| Discriminator learning rate | 0.002 | 0.002 | 1e-5 |
| target EMA decay rate $\mu$ | 0.9 | 0.9 | - |
| student EMA decay rate | 0.9999 | 0.9999 | - |
| N | 1 | 1 | 1 |
| ODE solver | Euler | Euler | Euler |
| Batch size | 128 | 2048 | 2 |
| Number of GPUs | 1 | 8 | 1 |
| Training iterations | 300K | 500K | 20k |

## A.2 MORE SAMPLES

Here we provide more samples in the appendix.

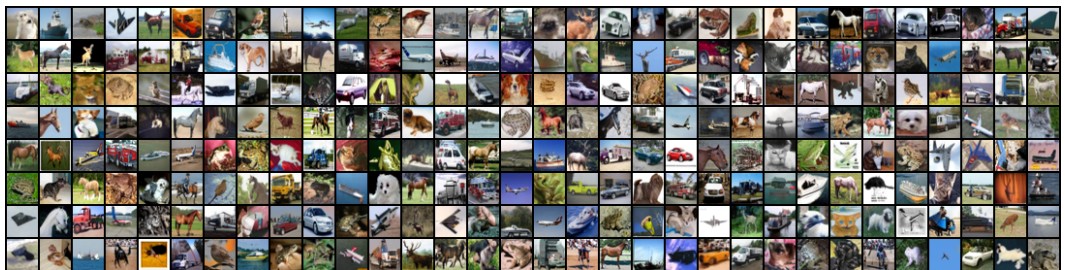

Figure 8: Samples generated by CCM (NFE=1) trained on CIFAR-10.

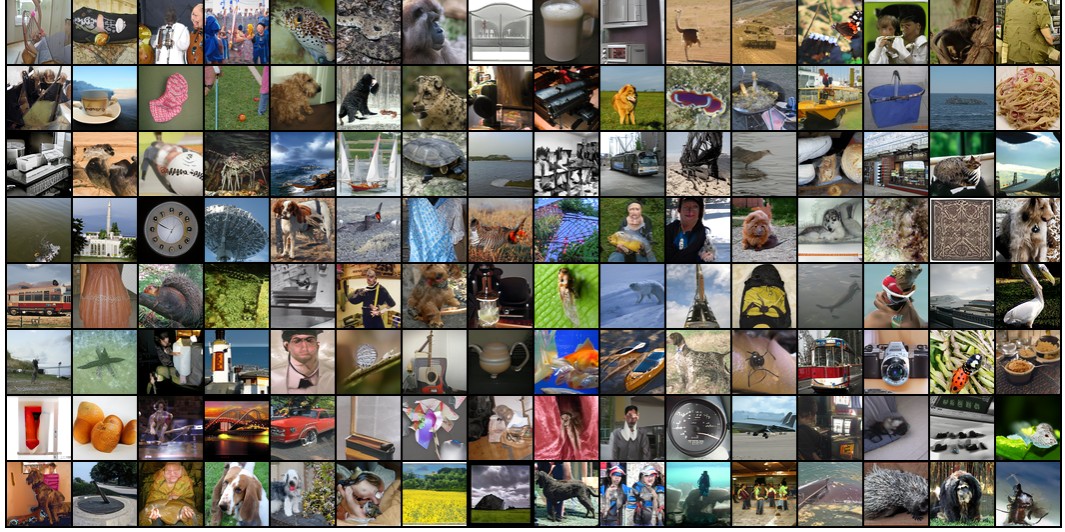

Figure 9: Samples generated by CCM (NFE=1) trained on ImageNet 64x64.