# OpenReview forum: "See Further When Clear: Adaptive Generative Modeling with Curriculum Consistency Model"
_ICLR.cc/2025/Conference — ICLR 2025 Conference Withdrawn Submission_

### Official Review · Reviewer_9tid · 2024-11-03

**Soundness:** 2
**Presentation:** 2
**Contribution:** 3
**Rating:** 5
**Confidence:** 4

**Summary:**

The paper introduces a straightforward and effective approach for optimizing the training time grid for discrete consistency models, resulting in competitive performance across various image generation benchmarks. By defining a notion of "learning complexity" using PSNR, the work provides insights on the training dynamics of consistency models.

**Strengths:**

S1 - The method is simple yet effective, demonstrating significant performance improvements over the base model in CIFAR and ImageNet settings.

S2 - The experiments are thorough, covering both Diffusion and Flow models and spanning unconditional, class-conditional, and text-conditional generation tasks.

**Weaknesses:**

W1 - The concept of "learning complexity" lacks theoretical justification. In Sec 3, the authors propose that high complexity leads to confusion, while low complexity reduces learning efficiency, but these observations appear mostly empirical. Including more theoretically motivated explanations, as seen in [1] regarding sampling time grid optimization, especially their discuss on the relation between time grid and the KL upper bound, would add depth to the discussion.

W2 - The authors should cite [2] and discuss its model, as it is closely related and also focuses on improving training efficiency in consistency models. Additionally, experiment results with NFE = 2 should be included for a more comprehensive comparison, as it's almost standard practice in most consistency model papers.

W3 - The notation $G_\theta$ is not clearly defined and is inconsistently used between Section 2.1 and Algorithm 1. In Section 2.1 the last variable in $G_\theta$ seems to be the target time step, while in Algorithm 1. it is replaced with condition $c$.

[1] Sabour, Amirmojtaba, Sanja Fidler, and Karsten Kreis. "Align your steps: Optimizing sampling schedules in diffusion models." arXiv preprint arXiv:2404.14507 (2024).
[2] Geng, Zhengyang, Ashwini Pokle, William Luo, Justin Lin, and J. Zico Kolter. "Consistency Models Made Easy." arXiv preprint arXiv:2406.14548 (2024).

**Questions:**

Given the potential for other similarity metrics, such as SSIM or LPIPS, to assess similarity in pixel space and thus learning complexity, have the authors explored these options?

---

### Official Review · Reviewer_BVFf · 2024-11-03

**Soundness:** 2
**Presentation:** 2
**Contribution:** 2
**Rating:** 3
**Confidence:** 5

**Summary:**

The paper proposes a novel training method for CMs that adaptively selects consequentive timesteps to ensure that the difficulty of the learning targets is maintained throughout the training process across different samples. The method is validated on CIFAR10, ImageNet64 and T2I generation, demonstrating strong performance when combined with adversatial training.

**Strengths:**

**S1 |** The paper addresses an important problem of improving the training of consistency models.

**S2 |** The proposed learning strategy is reasonable and well-motivated.

**S3 |** Compared with standard CD, CCM accelerates convergence and improves overall performance.

**Weaknesses:**

**W1 |** Recent works [1,2] have also addressed the training complexity and instabilities in CMs and proposed various related training techniques to improve convergence. However, these works are neither discussed nor compared with.

I believe the presence of [1,2] largely limits the main contribution and that CCM needs to be carefully compared against the techniques from these works, omitting the effect of the GAN loss.

**W2 |** CCM has been applied only to FM models on CIFAR10 and ImageNet64. It makes comparisons with the original CD, CTM, iCT[1], and ECM[2] difficult. Could the authors evaluate CCM on the corresponding diffusion models with and without GAN loss?

**W3 |** For T2I generation, the improvements seem negligible. FID, especially at 5K, is an unreliable metric, as noted in [3, 4]. Therefore, I believe the demonstrated FID gains may be unrepresentative or marginal. Could the authors conduct a human preference study or consider adding FID30K and alternative metrics, e.g., CMMD[3], ImageReward[5], and PickScore[6]?

**W4 |** Inconsistent terminology and notation regarding the student, teacher and target models/outputs. Figure 1 denotes a teacher model that maps $u$ to 1, yet in Section 2, the teacher maps  $t$ to $u$.

L210-211: $x_{est}$ - teacher output, $x_{target}$ - student output. L252: $x_{target}$ - teacher output. Algorithm 1: $x_{est}$ is a student output and $x_{target}$ is missing.

**W5 |** The related work section can be largely extended. I recommend citing and discussing CM-related works [1,2,7], as well as other distillation methods [8,9,10,11,12,13]. A comparison with these works would be highly beneficial.

**W6 |** The experiment in Figure 7 needs more details.

---
[1] Song et al. Improved Techniques for Training Consistency Models, 2023

[2] Geng et al. ECT: Consistency Models Made Easy, 2024

[3] Jayasumana et al. Rethinking FID: Towards a Better Evaluation Metric for Image Generation, 2024

[4] Podell et al. SDXL: Improving Latent Diffusion Models for High-Resolution Image Synthesis, 2023

[5] Kirstain et al. Pick-a-Pic: An Open Dataset of User Preferences for Text-to-Image Generation, 2023

[6] Xu et al. ImageReward: Learning and Evaluating Human Preferences for Text-to-Image Generation, 2023

[7] Salimans et al., Multistep Distillation of Diffusion Models via Moment Matching, 2024

[8] Berthelot et al. TRACT: Denoising Diffusion Models with Transitive Closure Time-Distillation, 2023

[9] Luo et al. Diff-Instruct: A Universal Approach for Transferring Knowledge From Pre-trained Diffusion Models, 2023

[10] Yin et al. One-step Diffusion with Distribution Matching Distillation 2023

[11] Yin et al. Improved Distribution Matching Distillation for Fast Image Synthesis, 2024

[12] Zhou et al. Score identity Distillation: Exponentially Fast Distillation of Pretrained Diffusion Models for One-Step Generation, 2024

[13] Kim et al. PaGoDA: Progressive Growing of a One-Step Generator from a Low-Resolution Diffusion Teacher, 2024

**Questions:**

**Q0 |** Please address the concerns and questions in Weaknesses.

**Q1 |** PSNR is computed between the student and teacher outputs. If I understand correctly, and the teacher output corresponds to the $u$->1 mapping, then PSNR and the CM loss are quite similar. Why not simply use the CM loss as is?

---

### Official Review · Reviewer_MYzj · 2024-11-04

**Soundness:** 4
**Presentation:** 3
**Contribution:** 2
**Rating:** 5
**Confidence:** 3

**Summary:**

The paper presents a novel training procedure for consistency models, introducing: 1) a dynamically adjusted training objective based on learning difficulty, measured by PSNR, and 2) a distillation target acquired through multi-step generation. The resulting model, named the Curriculum Consistency Model (CCM), demonstrates performance that is comparable to, if not better than, existing baseline models.

**Strengths:**

- The paper presents concrete motivation for each component of the proposed method.
- The proposed method shows some promising empirical results.
- The paper conducted various ablation studies to justify the importance of each proposed component.

**Weaknesses:**

- The dynamical adjustment, which is one of the major contribution of this paper, resembles **continuous-time training schedule** of ECT [1].
- Multi-step iterative generations will increase  the time complexity of training.
- Performance on CIFAR-10 is very promising, however, on large scale datasets such as ImageNet 64x64 or CoCo2017, the gain is either marginal or none. If the proposed method do increase time complexity of training, this result is not promising enough.


(Minor remarks)
- Personally, I believe Chapter 2 is not so kind for readers especially if they are unfamiliar with flow models.
- $G$ was not defined on line 151.
- The (0, 1) or (0, T) convention of FM and diffusion models seem to be opposite of one another, making it hard to read.
- Assigning $x_\text{target}$ is missing in Algorithm 1.


[1] Geng et al. "Consistency Models Made Easy"

**Questions:**

- In CIFAR-10 experiment, CCM is only applied to FM models. Was there a reason for not applying it to diffusion-based models?
- Prior works have reported that training of CM is difficult in terms of both stability and time complexity. Can you report those aspect of CCM as well?
- Have you tried *fine-tuning* a pretrained DM or FM with the proposed training procedure? According to ECT it reduces the required training time and increases stability of training.
- Does the proposed value of $T_\text{SNR}$ also generally work well with other models / datasets? If not, "our method is not very sensitive to PSNR" might not be a valid claim.

---

### Official Review · Reviewer_EGdj · 2024-11-04

**Soundness:** 2
**Presentation:** 1
**Contribution:** 2
**Rating:** 3
**Confidence:** 4

**Summary:**

The paper proposes the Curriculum Consistency Model (CCM), which stabilizes and balances the learning complexity across timesteps. It defines the distillation process as a curriculum and introduces Peak Signal-to-Noise Ratio (PSNR) as a metric to quantify the difficulty of each step in this curriculum. Extensive empirical studies are performed.

**Strengths:**

The strategy is rather simple and can easily be applied to diffusion models and flow-based models.

**Weaknesses:**

- The paper writing is disastrous. There are too many expression and grammar errors. In particular, the paper tile emphasizes consistency models but the flow-based models are highlighted in the main context.

-  The multi-step iterative generation method has been explored by the literature, see [1]. The GAN trick is not motivated.

- The paper should compare to more recent consistency models like improved CD, multi-step CM, etc. The arguments on the issues of CMs in this paper are partially problematic.

[1] SCott: Accelerating Diffusion Models with Stochastic Consistency Distillation.

**Questions:**

See above

---

### Note · Authors · 2024-11-14

I have read and agree with the venue's withdrawal policy on behalf of myself and my co-authors.